# ChemGymRL: An Interactive Framework for Reinforcement Learning for Digital Chemistry

**Chris Beeler**
University of Ottawa
Department of Mathematics and Statistics
Ottawa, Canada
christopher.beeler@uottawa.ca

**Sriram Ganapathi Subramanian**
University of Waterloo
Waterloo, Canada
Vector Institute for Artificial Intelligence
Toronto, Canada
s2ganapathisubrmanian@uwaterloo.ca

**Kyle Sprague**
University of Ottawa
Department of Mathematics and Statistics,
Ottawa, Canada
kspra093@uottawa.ca

**Mark Crowley**
Department of Electrical and Computer Engineering,
University of Waterloo,
Waterloo, Canada
mark.crowley@uwaterloo.ca

**Colin Bellinger**
National Research Council of Canada
Ottawa, Canada, and
Dalhousie University,
Faculty of Computer Science
Halifax, Canada
colin.bellinger@nrc-cnrc.gc.ca

**Isaac Tamblyn**
Department of Physics, University of Ottawa,
Ottawa, Canada, and
Vector Institute for Artificial Intelligence,
Toronto ON, Canada
isaac.tamblyn@uottawa.ca

## Abstract

This paper provides a simulated laboratory for making use of Reinforcement Learning (RL) for chemical discovery. Since RL is fairly data intensive, training agents 'on-the-fly' by taking actions in the real world is infeasible and possibly dangerous. Moreover, chemical processing and discovery involves challenges which are not commonly found in RL benchmarks and therefore offer a rich space to work in. We introduce a set of highly customizable and open-source RL environments, **ChemGymRL**, implementing the standard Gymnasium API. ChemGymRL supports a series of interconnected virtual chemical *benches* where RL agents can operate and train. The paper introduces and details each of these benches using well-known chemical reactions as illustrative examples, and trains a set of standard RL algorithms in each of these benches. Finally, discussion and comparison of the performances of several standard RL methods are provided in addition to a list of directions for future work as a vision for the further development and usage of ChemGymRL.

37th Conference on Neural Information Processing Systems (NeurIPS 2023).

# 1  Introduction

In Material Design, the goal is to determine a pathway of chemical and physical manipulation that can be performed on some starting materials or substances in order to transform them into a desired target material. Reinforcement Learning (RL) [29] is class of Machine Learning algorithms that learn by taking actions, making observations, viewing the results, and updating its model/or hypothesis/policy/beliefs. In other words, RL is a perfect analogy for the experimental scientist! The aim of this research is to demonstrate the potential of goal-based reinforcement learning (RL) in automated labs. Our experiments show that when given an objective (such as a target material) and a set of initial materials, RL can learn general pathways to achieve that objective. We postulate that well-trained RL chemist-agents could help reduce experimentation time and cost in these and related fields by learning to complete tasks that are repetitive, labour intensive and/or require a high degree of precision. To support this, we share the **ChemGymRL** environment that allows scientists and researchers to simulate chemical laboratories for the development of RL agents.

ChemGymRL is a collection of interconnected environments (or chemistry benches) that enable the training of RL agents for discovery and optimization of chemical synthesis. These environments are each a virtual variant of a chemistry "bench", an experiment or process that would otherwise be performed in real-world chemistry labs. As shown in Fig 1, the ChemGymRL environment includes reaction, distillation and extraction benches on which RL agents learn to perform actions and satisfy objectives.

The need for a simulated chemistry environment for designing, developing and evaluating artificial intelligence algorithms is motivated by the recent growth in research on topics, such as automated chemistry and self-driving laboratories [17, 16, 15, 27, 7], laboratory robots [12, 28, 3, 5, 25, 11, 1, 21] and digital chemistry for materials and drug discovery [14, 2, 20, 22, 13, 6, 32, 4, 24]. Given RL's adeptness at sequential decision making, and its ability to learn via online interactions with a physical or simulated environment without a supervised training signal, we see it as having a great potential within digital chemistry and self-driving laboratories. Recent work has demonstrated some successful applications of RL to automated chemistry [31, 33, 9]. Nonetheless, it remains an understudied area of research. Our work aims to partially addresses this problem by sharing an easy to use, extensible, open source, simulated chemical laboratory. This serves to simplify the design and development of application specific RL agents.

Although RL agents could be trained online in physical laboratories, this approach has many limitations, particularly in early stages of the research before mature policies exist. Training agents in a robotic laboratory in real-time would be costly, in both time and supplies, and restrictive, or even dangerous due to potential safety hazards. Our simulated ChemGymRL environment remedies this by allowing the early exploration phase to occur digitally, speeding up the process and reducing the waste of chemical materials. It provides a mechanism to design, develop, evaluate and refine RL for chemistry applications and researcher, which cannot safely be achieved in a physical setting.

The software is developed according to the `Gymnasium` standard, which facilitates easy experimentation and exploration with novel and off-the-self RL algorithms. When users download it, they gain access to a standard Gymnasium compatible environment that simulates chemical reactions using rate law differential equations, the mixing/settling of soluble and non-soluble solutions for solute extractions, the distillation of solutions, and a digital format for storing the state of the vessels used. In addition to this article, further detailed information about this software package, documentation and tutorials, including code and videos can be found at `urlsupressed`.

In our experimental results, we illustrate how to setup and use each bench with two distinct classes of reactions, along with how they can be extended to new reaction types. We evaluate the capabilities of a wide cross-section of off-the-shelf RL algorithms for goal-based policy learning in ChemGymRL, and compare these against hand-designed heuristic baseline agents. In our analysis, we find that only one RL off-the-shelf RL algorithm, Proximal Policy Optimization (PPO), is able to consistently outperform these heuristics on each bench. This suggests that the heuristics are a challenging baseline to compare to but that they are also far

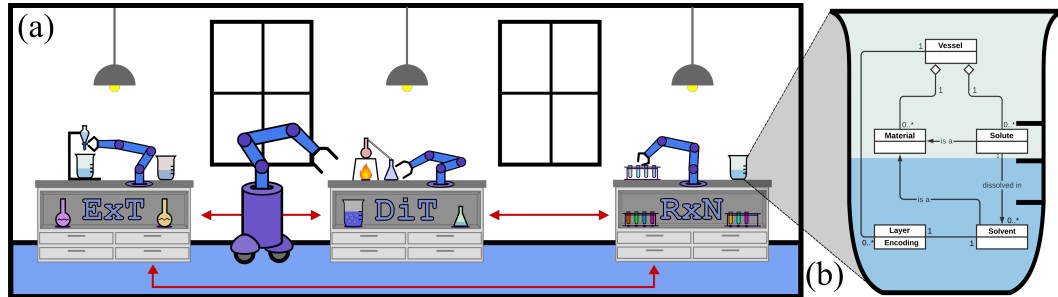

Figure 1: (a) The ChemGymRL simulation. Individual agents operate at each bench, working towards their own goals. The benches pictured are extraction (`ExT`), distillation (`DiT`), and reaction (`RxN`). The user determines which materials the bench agents have access to and what vessels they start with. Vessels can move between benches; the output of one bench becomes an input of another, just as in a real chemistry lab. Successfully making a material requires the skilled operation of each individual bench as well as using them as a collective. (b) Materials within a laboratory environment are stored and transported between benches within a vessel. Benches can act on these vessels by combining them, adding materials to them, allowing for a reaction to occur, observing them (thereby producing a measurement), etc.

from optimal. Thus there is space for an optimization approach such as RL to achieve optimal behavior on each bench. Near the end of the paper, we discuss some of the challenges, limitations and potential improvements in RL algorithms required to learn better, more sample efficient policies for discovery and optimization of material design pathways.

The remainder of the paper is organized as follows. The next section describes the Chem-GymRL environment, including the three primary benches: Reaction, Extraction and Distillation. Section 3 provides an overview of reinforcement learning and Section B contains a case study of the Wurtz Reaction and its use in each bench. Our experimental setup involves training off-the-shelf RL algorithms on each of the benches. The RL algorithms and hyperparameters are discussed in Section 4 and the specific laboratory settings and objectives used in our experiments are described in Section C. The results of the RL experiments are presented in Section 5 and the limitations of the simulations are discussed in Section 6 followed by our general conclusions and some ideas for future directions.

## 2 ChemGymRL

### 2.1 The Laboratory

The ChemGymRL environment can be thought of as a virtual chemistry laboratory consisting of different benches where a variety of tasks can be completed, see Fig. 1(a) for an overview.

The laboratory is comprised of 3 basic elements: **Vessels** contain materials, in pure or mixed form, and track their hidden internal state, **Shelves** are collections of vessels for input/output to benches, and **Benches** are simulations of particular chemistry activities

A bench must be able to receive a set of initial experimental supplies, possibly including vessels, and return the results of the intended experiment, also including modified vessels. Each bench has at least three common components: **Input**: target material given as a one-hot vector, **State**: vessels and contained materials, **Render**: Human Rendering and various possible numeric outputs for learning. In addition, every bench has it's own set of **Actions** and **Rewards**. The details and methods of how the benches interact with the vessels between these two points are completely up to the researcher using the framework, this includes the goal of the bench itself. In the following sections we describe the set of core benches we have define in the initial version of ChemGymRL.

### 2.2 The Benches

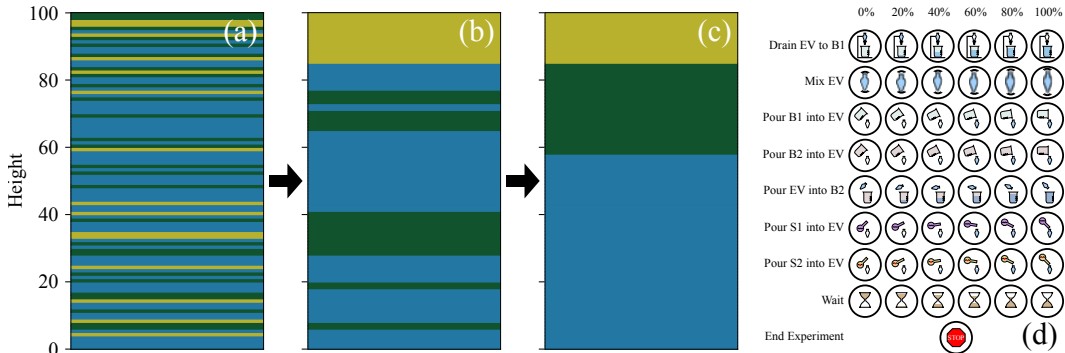

Figure 3: Typical observations seen in extraction bench (ExT) for a vessel containing air, hexane, and water. (a) The vessel in a fully mixed state. Each material is uniformly distributed throughout the vessel with little to no distinct layers formed. (b) The vessel in a partially mixed state. The air has formed a single layer at the top of the vessel and some distinct water and hexane layers have formed, however they are still mixed with each other. (c) The vessel in a fully settled state. Three distinct layers have formed in order of increasing density: water, hexane, and then air. (d) The icons representing each action and their multiplier values available in ExT. The extraction vessel (EV) is the primary vessel used, B1/B2 are the auxiliary vessels used in the experiment, and S1/S2 are the solvents available.

A short description of the four chemistry benches, see supplementary material for further details.

**Extraction Bench (ExT):** This bench provides methods to separate out undesired products from after a chemical reactions. It aims to isolate and extract certain dissolved materials from the input vessels. **Actions:** Transferring materials between different vessels and utilizing specifically selected solvents to separate materials from each other. **Rewards** is the difference in the relative purity of the desired solute at the first and final step of the episode.

**Distillation Bench (DiT):** This bench aims to isolate certain materials from an input vessel containing multiple materials. **Actions:** Transferring materials between a number of vessels and heating/cooling the vessel to separate materials from each other. **Rewards:** Amount and purity of target in output vessel.

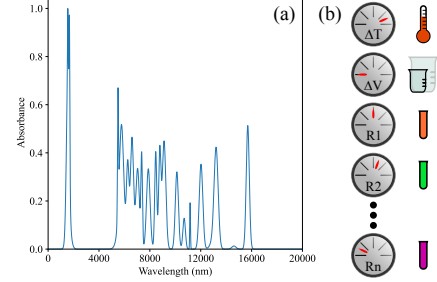

Figure 2: A visualization of the reaction bench (RxN) observation and action space. (a) An example of a UV-Vis spectra that would been seen in an observation and (b) The icons representing each action in RxN.

**Reaction Bench (RxN):** This bench enables the agent to transform available reactants into various products via a chemical reaction. **Actions:** The agent has the ability to control: the temperature of the vessel and the amounts of reactants added. **Rewards:** After the 20 steps have elapsed, the agent receives a reward equal to the molar amount of the target material produced.

The goal of the agent operating this bench is to modify the reaction parameters, in order to increase and/or decrease the yield of certain desired/undesired materials. The key to the agent's success in this bench is learning how best to allow certain reactions to occur such that the yield of the desired material is *maximized*, while the yield of the undesired material is *minimized*.

**Characterization Bench:** Not currently operated by an agent. Any observation of a vessel made by an agent must pass through this "bench". **Current Observations:** As visual layers, the same level of information as provided in human operated visualization (ordering of mixture layers).

# 3 Reinforcement Learning

Reinforcement Learning (RL) [30] can be used as a way to solve Markov Decision Processes (MDPs). MDPs are represented as a tuple $\langle S, A, R, T, \gamma \rangle$ where the $s \in S \subseteq \mathbb{R}^n$ denotes the state space, $a \in A \subseteq \mathbb{R}^m$ denotes the action space, $r \in R \subseteq \mathbb{R}$ denotes the reward function and $T = P(s_{t+1}|s_t, a_t)$ denotes the transition dynamics that provides the probability of state $s_{t+1}$ at the next time step given that the agent is in state $s_t$ and performs action $a_t$. The objective for an RL agent is to learn a policy $\pi(a|s)$ that maximizes the discounted sum of expected rewards provided by the equation $J_\pi(s) = \mathbb{E}_\pi[\sum_{t=0}^{\infty} \gamma^t r_t | s_0 = s]$, where $\gamma \in [0, 1)$ is the discount factor.

For this domain we look at two broad classes of RL algorithms, discrete action $Q$-learning approaches and discrete or continuous action Actor-Critic approaches. In $Q$-learning, a state-value function $Q$ is learned iteratively using the Bellman update $\mathcal{B}^* Q(s,a) = r(s,a) + \gamma \mathbb{E}_{s' \sim T(s'|s,a)}[\max_{a'} Q(s', a')]$. Here $s$ and $s'$ denote the current and next state respectively, $a$ and $a'$ denotes the current and next action respectively. An exact or approximate scheme of maximization is used to extract the greedy policy from the $Q$-function.

In actor-critic, the algorithm alternates between computing a value function $Q^\pi$ by a (partial) policy evaluation routine using the Bellman operator on the stochastic policy $\pi$, and then improving the current policy $\pi$ by updating it towards selecting actions that maximize the estimate maintained by the $Q$-values. This family of methods apply to both discrete and continuous action space environments so they can be used on any bench in chemistry gym environment.

# 4 Experimental Setup

To test and demonstrate the framework, we trained RL agents for 100K time steps across 10 environments in parallel (for a total of 1M time steps). Each experiment was limited to 256 time steps of experience (in each environment) to update policies/Value-functions. For the replay buffer, 1 millions experiences were used for DQN, SAC, and TD3. For exploration, the first 30K steps of DQN used a linear schedule from 1.0 down to 0.01, then fixed. All of these RL algorithms were performed using the Stable Baselines 3 [23] implementations.

In our discrete benches, Proximal Policy Optimization (PPO) [26], Advantageous Actor-Critic (A2C) [19] and Deep Q-Network (DQN) [18] were used. In our continuous benches, Soft Actor-Critic (SAC) [10] and Twin Delayed Deep Deterministic Policy Gradient (TD3) [8] were used instead of DQN. Note that we can choose between SAC and TD3 fairly arbitrarily since they are both off-policy algorithms that use Q-learning. The details of the methodology for each chemistry bench can be found in Appendix C.

## 4.1 Case Study

As a simple example, we outline how a particular chemical production process uses each of the benches.

We use the **Wurtz Reaction**, which are a very commonly used, and well-understood, approach for the formation of certain hydrocarbons. These reactions are of the form:

$$2\text{R-Cl} + 2\text{Na} \xrightarrow{\text{diethyl ether}} \text{R-R} + 2\text{NaCl}. \tag{1}$$

# 5 RL Results

## 5.1 Reaction Bench

Since reaction bench (RxN) has a continuous action space, we trained SAC and TD3 in addition to A2C and PPO. For the first experiment, we are looking at the Wurtz reaction dynamics. Given that we know the system dynamics in this case, we have also devised a heuristic agent for the experiment, which we expect to be optimal. This agent increases the temperature, and adds only the required reactants for the desired product immediately. This heuristic agent achieves an average return of approximately 0.62. Using the performance of

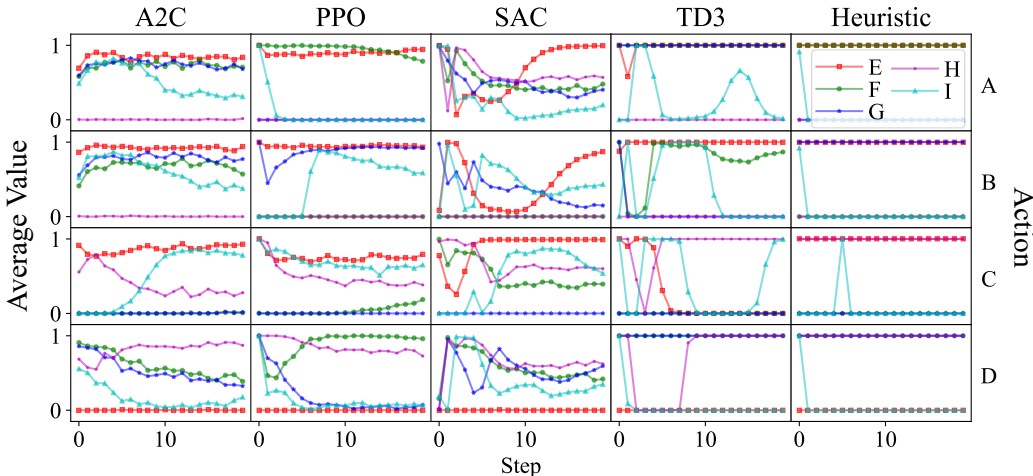

Figure 4: Fictitious `RxN`, average value of each action at every step for the best performing policies for each algorithm. The five curves in each box represents the sequence of actions for the five different target materials. The well performing policies are the ones that add only the required reactants (such as A2C and SAC), while the best performing policies are the ones that add them according to the right schedule (such as PPO).

this heuristic as a reference, the best and mean relative performances of the agents trained with each algorithm is shown in Fig. 8. Each algorithm can consistently give rise to agents that produce sodium chloride when requested. Since this is a by-product of all reactions in our set-up, it is the easiest product to create. While the other products are not hard to produce either, they require specific reactants, and in order to maximize the yield, they require the absence of other reactants. The PPO agents are able to match the heuristic agent for all targets, while some SAC and TD3 agents are able to come close on a few targets. A2C only comes close to the heuristic on producing sodium chloride.

The average return as a function of training steps for each algorithm is shown in Fig. 9. On average, the agents trained with each algorithm are able to achieve a return of at least 0.4. This is expected as even an agent taking random actions can achieve an average return of approximately 0.44. The agents trained with A2C, SAC, and TD3 do not perform much better than a random agent in most cases, however the ones trained with PPO significantly outperform it. While on average, A2C, SAC, and TD3 have similar performance, we saw in Fig. 8 that the best performing SAC and TD3 agents outperformed the best A2C agents.

The second `RxN` experiment uses reaction dynamics more complicated than the Wurtz reaction. In the Wurtz reaction, the agent need only add the required reactants for the desired product all together. In this new reaction, this is still true for some desired products, however not all of them. Similarly to the previous experiment, we also devised a heuristic agent for this experiment, which achieves an average return of approximately 0.83. Using the performance of the heuristic agent as reference again, the best and mean relative performances of the agents trained with each algorithm are shown in Fig. 10. Once again, PPO consistently produces agents that can match the performance of the heuristic agent. The best performing policies produced by A2C, SAC, and TD3 are able to nearly match the heuristic agent for all desired products excluding I. This is not unexpected as producing I requires producing intermediate products at different times during the reaction. On average, the policies produced by SAC and TD3 however, are unable to match the heuristic agent when asked to produce E. This is also not unexpected, given that producing E is penalized for all other desired products.

Unlike PPO, the other algorithms used appear to be less reliable at producing these best performing agents. This could be due to PPO learning these policies much faster than the other algorithms, as seen in Fig. 11. Since PPO converges to optimal behavior so quickly, there's very little room for variation in the policy. The other algorithms however are slowly

converging to non-optimal behaviors, leaving much more room for variation in the policies (and returns) that they converge to.

For the best performing agents produced by each algorithm, the average action values for each target are shown in Fig. 4. Looking at the heuristic policy, a constant action can be used for each target product, excluding I. When the target is I, the desired action must change after several steps have passed, meaning the agent cannot just rely on what the specified target is. Note that if all of a material has been added by step $t$, then it does not matter what value is specified for adding that material at step $t + 1$.

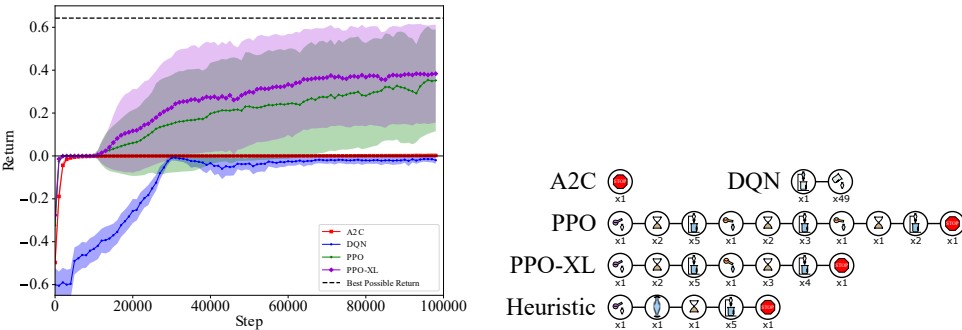

Figure 5: **Left:** Wurtz `ExT`, average return with $\sigma$ shaded. The PPO and PPO-XL agents consistently acquire positive returns. The A2C agents learn policies which perform equivalently to ending the experiment immediately and are unable to escape those local maxima. The DQN agents acquire negative return, which is a worse performance than not running the experiment. **Right:** The sequence of actions with the highest return when dodecane is the target material seen during the rollout of the best performing policy learned by each algorithm. Each picture represents an action and average value described by Fig. 3(d). The number beneath the image represents how many times that action was repeated.

The best performing agent for each algorithm were all able to produce E when requested and Fig. 4 shows that they each have learned to add A, B, C, and not D. It can also be seen that all four algorithms learned to add two of A, B, or C in addition to D, then add the third one several steps later when I is the target product, mimicking the behavior of the heuristic policy. Note that even though the heuristic waits to add C, waiting to add A or B instead would be equally optimal. While each algorithm does this, PPO and A2C do so better than the others. PPO is also the only one that succeeds in both of these cases, showing that an RL agent can learn the required behavior in this system.

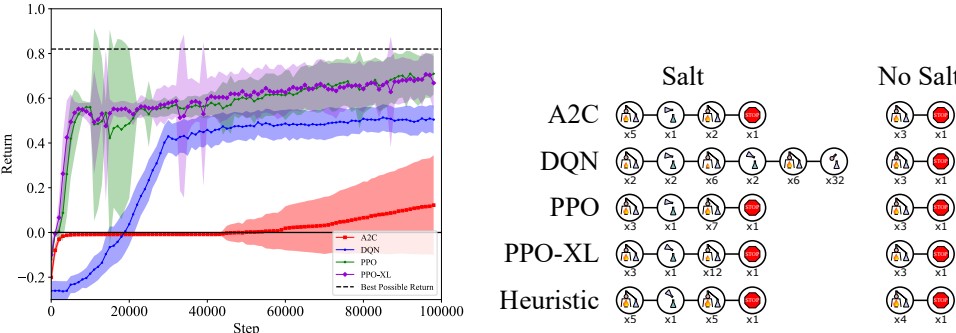

Figure 6: **Left:** Wurtz `DiT`. The DQN, PPO, and PPO-XL agents consistently acquire positive returns whereas the A2C agents only get positive returns on average. The PPO-XL policies outperform the other algorithms both on average and in the best case scenarios. **Right:** The sequences of actions with the highest return produced by the best performing policy learned with each algorithm for two cases: when salt is and is not initially present in the distillation vessel with another material.

## 5.2 Extraction Bench

With the results seen in the RxN tests, we now move onto the extraction bench (ExT) experiment. Regardless of the target material in our Wurtz extraction experiment, the optimal behavior is quite similar so we will not focus on the different cases as before. Since the ExT uses discrete actions, we replace SAC and TD3 with DQN. We also use what we call PPO-XL which is PPO trained with more environments in parallel. Unlike the reaction bench, we do not have an analytic solution for this bench, therefore we have devised a heuristic policy for this experiment based on what an undergraduate chemist would learn. However, as the dynamics are more complex we do not necessarily expect it to be optimal.

As seen in Fig. 5, the agents trained with A2C do not achieve a return above zero, while the agents trained with DQN ended up achieving a negative return. Not only do both PPO and PPO-XL produce agents that achieve significantly more reward than the other algorithms, they are able to outperform the heuristic policy as well. On average, the best performing agent trained with PPO-XL manages to achieve a return of approximately 0.1 higher than the heuristic (see Fig. 5), resulting in roughly a 10% higher solute purity. PPO and PPO-XL consistently outperform the agents trained with the other algorithms.

As shown in Fig. 5(b), the action sequences of the policies learned from A2C, DQN, and PPO are quite different. The action sequences of the policies learned by PPO and PPO-XL are much more similar, as expected. The first half of these sequences are comparable to the heuristic, however the agents in both cases have learned a second component to the trajectory to achieve that extra return. Interestingly, both PPO and PPO-XL agents have learned to end the experiment when they achieve the desired results, whereas the A2C and DQN agents do not. PPO once again shows that an RL agent can learn the required behavior in this system.

## 5.3 Distillation Bench

Lastly, we now move onto the final experiment, distillation bench (DiT). Similar to ExT, the desired target material in the Wurtz distillation experiment does not have much effect on the optimal behavior so we will not focus on the different target cases. Instead we will focus on the different cases of when salt is and is not present with another material in the initial distillation vessel. Note that a single agent operates on both of these cases, not two agents trained independently on each case. As before, we have devised a heuristic policy and as with the RxN experiments, we expect it to be optimal once again. In Fig. 6 we can see that on average, the algorithms (excluding A2C) converge faster than in the other experiments, however there is much less variation in return compared to before.

For the case when salt and an additional material are present, the best performing agents trained with PPO and PPO-XL modify their actions similar to the heuristic policy, achieving the optimal return in both cases. The best performing agent trained with A2C modifies their actions in a similar fashion, however it does so in a way that also achieves a much lower return. The best performing agent trained with DQN makes much more significant changes to their policy, however it still achieves a return closer to optimal than A2C. This shows that the expected behavior in our final bench can also be learned by an RL agent.

## 6 Limitations

ChemGymRL has limitations; any reaction or material that one wishes to model must be predefined with all properties specified by the user. Additionally, the solvent dynamics are modeled using simple approximations and while they suffice for these introductory tests, they would not for real-world chemistry.

As previously mentioned, the ChemGymRL framework was designed in a modular fashion for the ease of improvement. The differential equations used to model the reactions could be replaced with a molecular dynamics simulation. This would allow RxN to operate with on a more generalizable rule-set. Without having to manually define the possible reactions, the RxN could be used to discover new, more efficient reaction pathways by an RL agent. Currently, the reward metric used in RxN is the molar amount of desired material produced

by the agent. If this metric was changed to reflect a certain desired property for the produced material, then the `RxN` could be used for drug discovery. Making similar improvements to `ExT` and `DiT`, the RL agent could then learn to purify these new discoveries.

## 7 Conclusions and Future Work

We have introduced and outlined the ChemGymRL interactive framework for RL in chemistry. We have included three benches that RL agents can operate and learn in. We also include a characterization bench for making observations and presented directions for improvement. To show these benches are operational, we have successfully, and reproducibly, trained at least one RL agent on each of them. Included in this framework is a vessel state format compatible with each bench, therefore allowing the outputs of one bench to be the input to another.

As future work, the lab manager environment will be formatted in a way that allows an RL agent to operate in it. We would also like to try other kinds of reactions and more classes of RL methods using ChemgymRL.

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

# A  Appendix : Bench Details

## A.1  Reaction Bench (RxN)

The sole purpose of the **reaction bench (RxN)** is to allow the agent to transform available reactants into various products via a chemical reaction. The agent has the ability to control temperature and pressure of the vessel as well as the amounts of reactants added. The mechanics of this bench are quite simple in comparison to real-life, which enables low computational cost for RL training. Reactions are modelled by solving a system of differential equations which define the rates of changes in concentration (See Appendix).

The goal of the agent operating on this bench is to modify the reaction parameters, in order to increase, or decrease, the yield of certain desired, or undesired, materials. The key to the agent's success in this bench is learning how best to allow certain reactions to occur such that the yield of the desired material is maximized and the yield of the undesired material is minimized. Therefore the reward in this bench is zero at all steps except the final step, at which point it is the difference in the number of mols of the desired material and undesired material(s) produced.

***Observation Space:*** In this bench, the agent is able to observe a UV-Vis absorption spectra of the materials present in the vessel as shown in Fig. 2(a), the normalized temperature, volume, pressure, and available materials for the system.

***Action Space:*** The agent can increase or decrease the temperature and volume of the vessel, as well as add any fraction of the remaining reactants available to it. In this bench, the actions returned by an agent are a continuous valued vector of size $n + 2$, where $n$ is the number of reactants. These actions are also shown in Fig. 2(b).

A main feature of ChemGymRL is its modularity. If one wanted to make the results of RxN more accurate and generalizable, they could replace the current system of differential equations with a molecular dynamics simulation without needing to change how the agent interacts with the bench or how the bench interacts with the rest of ChemGymRL.

## A.2  Extraction Bench (ExT)

Chemical reactions commonly result in a mixture of desired and undesired products. Extraction is a method to separate them. The extraction bench (ExT) aims to isolate and extract certain dissolved materials from an input vessel containing multiple materials through the use of various insoluble solvents. This is done by means of transferring materials between a number of vessels and utilizing specifically selected solvents to separate materials from each other.

A simple extraction experiment example is extracting salt from an oil solvent using water. Suppose we have a vessel containing sodium chloride dissolved in hexane. Water is added to a vessel and the vessel is shaken to mix the two solvents. When the contents of the vessel settle, the water and hexane will have separated into two different layers.

***Observation Space:*** For a visual representation of the solvent layers in the vessel for the agent, as seen in Fig. 3(a)-(c), we sample each solvent corresponding to each pixel using the relative heights of these distributions as probabilities. This representation makes this bench a partially observable Markov decision process (POMDP). The true state is not observed because the observations do not show the amount of dissolved solutes present nor their distribution throughout the solvents. In this set up, very light solvents will quickly settle at the top of the vessel, while very dense solvents will quickly settle at the bottom. The more similar two solvents densities are, the longer they will take to fully separate.

***Action Space:*** The agent here has the ability to mix the vessel or let it settle, add various solvents to the vessel, drain the contents of the vessel into an auxiliary vessel bottom first, pour the contents of the vessel into a secondary auxiliary vessel, and pour the contents of either auxiliary vessel into each other or back into the original vessel. Unlike in RxN, only one of these actions can be performed at a time, therefore the actions returned by the agent conceptually consist of two discrete values. The first value determines *which* of the processes are performed and the second value determines the magnitude of that process. If the drain

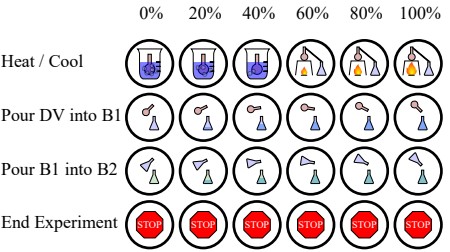

Figure 7: The icons representing each action and their multiplier values available in `DiT`. The distillation vessel (DV) is the primary vessel and B1/B2 are the auxiliary vessels in the experiment.

action is selected by the first value, then the second value determines how much is drained form the vessel. Including the ability to end the experiment, the agent has access to 8 actions with 5 action multiplier values each. These actions are depicted in Fig. 3(d). Practically however, the actions returned by the agent consist of a single discrete values to reduce redundancy in the action space.

The goal of the agent in this bench is to use these processes in order to maximize the purity of a desired *solute* relative to other *solutes* in the vessel. This means the agent must isolate the desired solute in one vessel, while separating any other solutes into the other vessels. Note that the solute's relative purity *is not* affected by the presence of solvents, only the presence of other solutes. Therefore the reward in this bench is zero at all steps except the final step, at which point it is the difference in the relative purity of the desired solute at the first and final steps.

### A.3  Distillation Bench (`DiT`)

Similar to the `ExT`, the distillation bench (`DiT`) aims to isolate certain materials from a provided vessel containing multiple materials (albeit with a different process). This is done by means of transferring materials between a number of vessels and heating/cooling the vessel to separate materials from each other.

A simple distillation example is extracting a solute dissolved in a single solvent. Suppose we have a vessel containing sodium chloride dissolved in water. If we heat the vessel to $100°$C, the water will begin to boil. With any added heat, more water will evaporate and be collected in an auxiliary vessel, leaving the dissolved sodium chloride behind to precipitate out as solid sodium chloride in the original vessel.

*Observation Space:* For a visual representation for the agent, we use the same approach described for `ExT`. For the precipitation of any solutes, we define a precipitation reaction and use the same approach described for `RxN`.

*Action Space:* The agent has the ability to heat the vessel or let it cool down and pour the contents of any of the vessels (original and auxiliaries) into one another. When the agent heats/cools the vessel, the temperature of the vessel and its materials are altered by a well-understood formula. Similar to the `ExT` bench, only one of these processes (heating/cooling) can be done at a time. Therefore in the `DiT` bench, the action returned by the agent again can be seen as two-part with an indicator of the action and a magnitude component. With the inclusion of the ability to end the experiment, the agent then has access to 4 actions, each with 10 action multiplier values. These actions are depicted in Fig. 7. Just as in `ExT`, the actions actually returned by the agent are flattened into a single discrete value to reduce redundancy in the action space.

The goal of the agent in this bench is to use these processes to maximize the absolute purity of a desired material in the vessel. This means the agent must isolate the desired *material* in one vessel, while separating any other *materials* into other vessels.

### A.4 Characterization Bench

The characterization bench is the primary method to obtain insight as to what the vessel contains. The purpose of the characterization bench is not to manipulate the input vessel, but to subject it to analysis techniques that observe the state of the vessel, possibly including the materials inside it and their relative quantities. This allows an agent or user to observe vessels, determine their contents, and allocate the vessel to the necessary bench for further experimentation.

The characterization bench is the only bench that is not "operated". A vessel is provided to the bench along with a characterization method and the results of said method on that vessel are returned. Currently, the characterization bench consists of a UV-Vis spectrometer that returns the UV-Vis absorption spectrum of the provided vessel. Each material in ChemGymRL has a set of UV-Vis absorption peaks defined and the UV-Vis spectrum for a vessel is the combination of the peaks for all materials present, weighted proportionally by their concentrations. In future versions of ChemGymRL we will expand the characterization bench to include other forms of partial observation.

## B Appendix: Wurtz Reaction Case Study

As a simple example, we outline how a particular chemical production process uses each of the benches.

We use the **Wurtz Reaction**, which are a very commonly used, and well-understood, approach for the formation of certain hydrocarbons. These reactions are of the form:

$$2\text{R-Cl} + 2\text{Na} \xrightarrow{\text{diethyl ether}} \text{R-R} + 2\text{NaCl}. \tag{2}$$

## C Chemistry Bench Methodology

### C.1 Reaction Bench Methodology

For the reaction bench (RxN), we consider two chemical processes. In both processes, each episode begins with a vessel containing 4 mols of diethyl ether, and operates for 20 steps. We chose 20 steps because it's long enough that the agent can explore the space to find the optimal behavior but short enough that the reward acquired at the end of the episode can be propagated back through the trajectory. In the first process, the agent has access to 1.0 mol each of 1, 2, 3-chlorohexane, and sodium, where the system dynamics are defined by the Wurtz reaction outlined above. Each episode, a target material is specified to the agent via length 7 one-hot vector where the first 6 indices represent the 6 Wurtz reaction products and the last represents NaCl. After the 20 steps have elapsed, the agent receives a reward equal to the molar amount of the target material produced.

In the second experiment, we introduce a new set of reaction dynamics given by

$$\begin{aligned} A + B + C &\to E \\ A + D &\to F \\ B + D &\to G \\ C + D &\to H \\ F + G + H &\to I \end{aligned} \tag{3}$$

where the agent has access to 1.0 mol of $A$, $B$, $C$ and 3.0 mol of $D$. We introduce this second reaction explore different mechanics required in the optimal solution. This reaction includes undesired and intermediate products, adding difficulty to the problem. Each episode, a target material is specified to the agent via length 5 one-hot vector with indices representing $E$, $F$, $G$, $H$, and $I$. If the target is $E$, the agent receives a reward equal to the molar amount of $E$ produced after the 20 steps have elapsed. Otherwise, the agent receives a reward equal to the difference in molar amounts between the target material and $E$ after the 20 steps have elapsed. Here, $E$ is an undesired material. The reaction $A + B + C \to E$ occurs quickly relative to the others, adding difficulty to the reaction when $E$ is not the target.

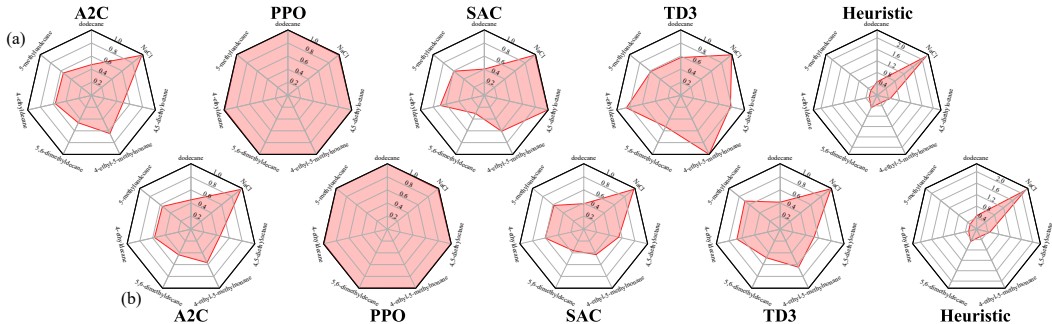

Figure 8: Radar graphs detailing the average return of each policy with respect to each target material in Wurtz RxN. Panel (a) uses the best policy produced from 10 runs, whereas panel (b) averages across the 10 runs (still using the best policy of each run). Returns of each RL algorithm are relative to the heuristic policy and clipped into the range $[0, \infty)$. Here, the PPO agents consistently outperform the A2C, SAC, and TD3 agents for all 7 target materials. Target materials with high returns across each algorithm (such as sodium chloride) appear to be easier tasks to learn, where target materials with less consistent returns across each algorithm (such as 5,6-dimethyldecane) appear to be more difficult tasks to learn.

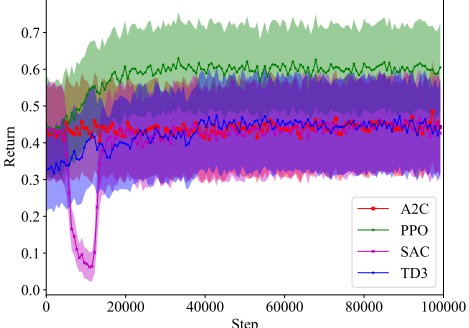

Figure 9: Wurtz RxN, average return with $\sigma/5$ shaded, 10 runs for each algorithm with 10 environments in parallel per run, 1M (100K sequential steps x 10 environments) total steps per run, averages are over 3200 returns. The performance of each algorithm converges before 300K total steps, with only PPO converging on an optimal policy. Despite training for an additional 700K total steps, A2C, SAC, and TD3 were not able to escape the local maxima they converged to.

## C.2 Extraction Bench Methodology

For the extraction bench (ExT), we consider a layer-separation process where the agent operates for up to 50 steps. We chose a larger number of steps in this experiment because the optimal solution is more complicated than the previous bench. Similar to the Wurtz reaction, the target material is specified via length 7 one-hot vector. Each episode begins with a vessel containing 4 mols of diethyl ether, 1 mol of dissolved sodium chloride, and 1 mol of one of the 6 Wurtz reaction products. The Wurtz reaction product contained in the vessel is the same as the target material, unless the target material is sodium chloride, in which case dodecane is added since sodium chloride is already present. After the episode has ended, the agent receives a reward equal to the change in solute purity of the target material weighted by the molar amount of that target material, where the change in solute purity is relative to the start of the experiment. If the target material is present in multiple vessels, a weighted average of the solute purity across each vessel is used.

As an example, consider when the target material is dodecane. In this experiment, the 1 mol of dissolved sodium chloride becomes 1 mol each of $Na^+$ and $Cl^-$, so the initial solute purity of dodecane is 1/3. Suppose we end the experiment with 0.7 mols of dodecane with

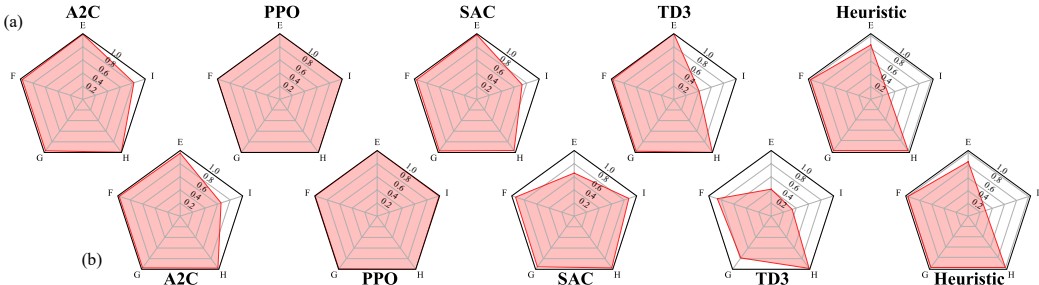

Figure 10: Radar Graph detailing the average return of each policy with respect to each target material in Fictitious RxN. Panel a) uses the best policy produced from 10 runs, whereas panel b) averages across the 10 runs (still using the best policy of each run). Returns of each RL algorithm are relative to the heuristic policy and clipped into the range $[0, \infty)$. Again, the PPO agents consistently outperform the A2C, SAC, and TD3 agents for all 5 target materials, however it is not as significant of a gap as in Wurtz RxN. Target materials with high returns across each algorithm (such as F, G, and H) appear to be easier tasks to learn, where target materials with less consistent return across each algorithm (such as E and I) appear to be more difficult tasks to learn.

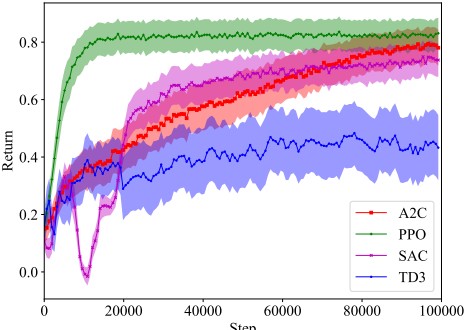

Figure 11: Fictitious RxN, average return with $\sigma/5$ shaded, 10 runs for each algorithm with 10 environments in parallel per run, 1M (100K sequential steps x 10 environments) total steps per run, averages are over 3200 returns. PPO quickly converges to an optimal policy, like in Wurtz RxN. Unlike in Wurtz RxN, the other algorithms take much longer to converge. While they still converge to sub-optimal performances, the gap between optimal performance is less severe.

0.2 mols each of $Na^+$ and $Cl^-$ in one vessel, and the remaining molar amounts in a second vessel. Dodecane has a solute purity of 7/11 and 3/19 in each vessel respectively. The final solute purity of dodecane would be $0.7 \times 7/11 + 0.3 \times 3/19 \approx 0.493$. Thus the agent would receive a reward of $0.159$.

### C.3 Distillation Bench Methodology

For the distillation bench (DiT), we consider a similar experimental set-up to the ExT one. Each episode begins with a vessel containing 4 mols of diethyl ether, 1 mol of the dissolved target material, and possibly 1 mol of another material. If the target material is sodium chloride, the additional material is dodecane, otherwise the additional material is sodium chloride. After the episode has ended, the agent receives a reward calculated similarly to the ExT, except using absolute purity rather than solute purity.

