# OpenReview forum: "ChemGymRL: An Interactive Framework for Reinforcement Learning for Digital Chemistry"
_NeurIPS.cc/2023/Workshop/AI4Science — NeurIPS2023-AI4Science Poster_

### Official Review · Reviewer_TiqV · 2023-10-20
**Potentially highly useful RL environment**

**Rating:** 7
**Confidence:** 4

**Review:**

This paper provides a Gymnasium-like interactive environment, ChemGymRL,  for training RL algorithms under a variety of chemistry tasks. The framework is flexible and modular, allowing for the incorporation of a variety of RL algorithms, chemicals, rewards, and processes (distillation, extraction, reaction, etc). Overall the framework is exciting as it will potentially be used to replace real-life chemistry experiments. I recommend it to be accepted to the workshop.

Pros
1. The paper is well written, with a clear description of the framework as well as solid simulation results to validate the framework.
2. The framework is flexible and may allow for easy plug-in use under a variety of RL-related benchmarks, including algorithms for molecular dynamics (MD).
3. The framework may replace some real-life experiments or at least become a realistic benchmark for algorithms involving chemical reactions, thereby reducing the cost and time needed for performing wet-lab experiments.

Cons
1. Since ChemGymRL is an RL simulation environment, it is important to assess its usability through code documentation and demos. The current manuscript does not include much of it. It would be helpful to include a wiki/git page for a step-by-step instruction on how to run the framework with customized components.
2. Even though the environment can be used to replace some real-life experiments, it still remains unclear to me what specific experiments will benefit substantially through using this environment. For simple reactions such as the Wurtz Reaction demonstrated in the paper, there seems to be no pressing need to train an RL agent as the optimal policy is already known.  I'm curious about which specific problem in chemistry that involves RL will ChemGymRL facilitate. For example, RL is widely applied in de novo drug design. Can ChemGymRL be used to train/evaluate RL algorithms that generate novel protein structures? How to do it (which bench to use? which modules need to be modified?)?

---

### Official Review · Reviewer_qibx · 2023-10-25
**Authors have proposed an RL environment called ChemGymRL for chemical reaction, and thus ChemGymRL can be used to learn RL policy for virtual synthesis of target materials.**

**Rating:** 7
**Confidence:** 4

**Review:**

In this work authors have developed an environment called ChemGymRL for chemical reaction, which can used to train RL agent and learn policy for the synthesis condition of target material from input reactant. The paper is well written and easy to follow and will be useful for researchers to test RL algorithms for chemical synthesis. However, their are one area where this paper needs improvement as explained below:
1. The author needs to provide more details regarding the dynamics of their environment design of ChemGymRL. More specifically what assumptions/approximation their environment have for different chemical reactions and chemical reaction dynamics mismatch between ChemGymRL and real world condition used for chemical synthesis currently supported in their  environment.
2. Authors have added a line in the abstract on page 1 line2-3  "Since RL is fairly data intensive, training agents ‘on-the-fly’ by taking actions in the real world is infeasible and possibly dangerous." However, I don't see anything related what kind of safety their environment (ChemGymRL) and used RL algorithm   provides towards risky exploratory or infeasible actions. For example: on page 5 line 179 it is written "This agent increases the temperature, and adds only the required reactants for the desired product immediately". Different materials have different allowed maximum temperature to which they should be heated and the period of time. What kind of safety  ChemGymRL provides to make sure agent does not allow risky action (say heating beyond max allowed temperature).
3. If agent purposes an optimal  synthetic pathway for a material but after preformation say either Molecular dynamics simulation or real experiment turns out that pathway have low performance. In this case how this information will be incorporated within  ChemGymRL for policy improvement for that material.

---

### Meta-Review · Area_Chair_SLjH · 2023-10-26

**Recommendation:** Accept (Poster)
**Confidence:** 4

**Metareview:**

The paper introduces a novel gym RL environment. Reviewers voted for acceptance on the ground of its potential to help the effort of digitization of chemistry and the overal high quality of the paper. The work would benefit from elucidating what open challenges does the enviornment helps solve by enabling training or benchmarking models for. As one reviewer remarked, implemented reactions or procedures do seem quite simple to learn. Thank you for your submission and I am happy to recommend acceptance of the paper.